# Antibiotic Prophylaxis for Hepato-Biliopancreatic Surgery—A Systematic Review

**DOI:** 10.3390/antibiotics11020194

**Published:** 2022-02-01

**Authors:** Francesca Steccanella, Paolo Amoretti, Maria Rachele Barbieri, Fabio Bellomo, Alessandro Puzziello

**Affiliations:** General Surgery Unit, AOU San Giovanni di Dio e Ruggi d’Aragona, University of Salerno, Largo Città di Ippocrate, 84131 Salerno, Italy; fra.steccanella@gmail.com (F.S.); paoloamoretti95@gmail.com (P.A.); rachelemaria92@gmail.com (M.R.B.); fabiobellomo10@gmail.com (F.B.)

**Keywords:** antibiotic prophylaxis, hepato-biliopancreatic surgery, surgical site infections

## Abstract

Background: Surgical site infections (SSIs) are among the most important determinants of morbidity after HBP surgery. Their frequency after HPB surgery is variable, from 1–2% after elective cholecystectomy to 25% after PD. Methods: A systematic review was performed to assess the role of antimicrobial prophylaxis (AP) in HPB elective surgery. Articles published between 2015 and 2021 were obtained; those before 2015 were not included because they antedate the WHO guidelines on SSI prevention. We conducted three different research methods for liver resection, elective cholecystectomy and pancreatic and biliary surgery regarding patients requiring preoperative biliary drainage. Results: Hepatic surgery, improvement in surgical technique and perioperative management lead to a very low SSI. One preoperative 2 g cefazolin dose may be adequate for surgical prophylaxis. From preoperative biliary drainage, we can derive that patients’ homeostasis rather than AP plays a paramount role in reducing postoperative morbidity. The time from biliary drainage could be an essential element in decision making for surgical prophylaxis. In the case of low-risk cholecystectomy, it is not easy to draw definitive conclusions about the effect of AP. Data from the literature are inconsistent, and some risk factors cannot be predicted before surgery. Conclusion: in our opinion, a strict preoperative cefazolin dose strategy can be reasonable in HBP surgery until a large-scale, multicentric RCT brings definitive conclusions.

## 1. Introduction

Hepato-biliopancreatic (HPB) surgery includes liver, pancreas, bile duct, and gallbladder surgery. It represents a heterogeneous field, ranging from simple cholecystectomy to complex pancreatic and liver oncological resection, performed both laparoscopically and with the open technique.

Surgical site infections (SSIs) are one of the most important determinants of morbidity after hepatobiliary surgery. Their frequency after HPB surgery is highly variable, depending on the complexity of the procedure and intrinsic characteristics of the patient. It can vary from 1–2% after elective cholecystectomy to 25% after pancreatoduodenectomy (PD) [1,2]. According to CDC wound classification [3], HPB surgery is classified as clean-contaminated because the bile duct is transected and, for some procedures, the digestive tract is entered. In these patients, parenteral antimicrobial prophylaxis is administered to prevent SSI and SSI-related morbidity and mortality.

Current surgical prophylaxis guidelines [4] were published in 2013. As a general rule, primary surgical antimicrobial prophylaxis should be administered, when indicated, within one hour before the incision, followed by additional injections if the duration of the operation exceeds two half-lives of the drug. The administration should not be extended beyond 24 h after operation and, in most cases, a single preoperative dose is recommended.

Pancreaticoduodenectomy is considered a gastroduodenal procedure with a grade A recommendation of a cefazolin single preoperative dose. Cholecystectomy, exploration of the common bile duct and choledocho-enterostomy are grouped as biliary tract procedures. In these cases, the recommendations are a single dose of cefazolin for open biliary tract operation and no prophylaxis in low-risk laparoscopic cholecystectomy because of the very low rate of SSI. These indications were confirmed in WHO guidelines on surgical site infection prevention in 2015. This paper defines surgical prophylaxis as a single antimicrobial dose administered within 120 min [5].

We think that, however, at least three issues remained unsolved in current guidelines [4,5].

A first hepatic resection without bile duct reconstruction is not explicitly addressed. With recent improvements in perioperative management and surgical techniques, hepatectomy has become a safer surgical procedure [6]. Moreover, surgical indications for hepatic lesions have more effective multimodal oncological treatments [7,8].

The second issue is bacterobilia in patients with preoperative jaundice. Bile is usually sterile, but preoperative drainage because of jaundice increases the risk of bacterobilia. This is a matter of concern mainly for pancreatic head carcinoma and extrahepatic cholangiocarcinoma. Bacterobilia following bile duct intervention is polymicrobial and increases antimicrobial resistance; it is also a risk factor for SSIs [8]. Current guidelines include pancreaticoduodenectomy in gastroduodenal procedures [4], and it remains unclear whether bile culture results can lead to a change in the management of antimicrobial prophylaxis.

Last but not least is antimicrobial prophylaxis in low-risk elective cholecystectomy. Current guidelines relied on studies often underpowered, varied in the control groups used, type and duration of prophylaxis, and in SSI definition and reporting. Moreover, some of the known risk factors for SSI cannot be determined before surgical intervention. Recommendations are thus almost ambiguous, stating that prophylaxis is not recommended in low-risk elective laparoscopic cholecystectomy, but also that it may be reasonable to give a single dose of antimicrobial to all patients undergoing cholecystectomy [4].

In this article, we revise the literature regarding antimicrobial prophylaxis in HPB surgery following WHO global guidelines publications [5]. We focus on three previously discussed issues to look for some updates in more recent publications.

## 2. Materials and Methods

A systematic review of the literature published in peer-reviewed journals was performed to assess the role of antimicrobial prophylaxis in HPB elective surgery. Articles published in English between 2015 and 2021 were retained. Studies published before 2015 were not included in this review because they preceded the WHO guidelines on SSI prevention [7].

We conducted three different reviews for liver resection, elective cholecystectomy, and pancreatic and biliary surgery regarding patients requiring preoperative biliary drainage.

The search strategy used a prospectively defined algorithm in PubMed and the Cochrane Database of Systematic Reviews and was conducted on 1 October 2021. Mesh terms included: antimicrobial prophylaxis AND prophylaxis, antimicrobial. Mesh terms were matched with the following keywords: (“liver resection” OR “hepatectomy” OR “metastasectomy” OR “liver surgery”); (“cholecystectomy” OR “laparoscopic cholecystectomy” OR “gallbladder”); (“pancreatectomy” OR “pancreatoduodenectomy” OR “distal pancreatectomy” OR “pancreatic surgery”). A manual search of the reference list from relevant articles was also carried out.

Inclusion criteria were (A) clinical trials and meta-analyses evaluating primary surgical prophylaxis with parenteral antibiotics (A) in gallbladder, liver, and pancreatic surgery (C) from 2015 to 2021.

Exclusion criteria were studies dealing with non-oral prophylaxis, decolonization, or not reporting outcome measures.

## 3. Results

### 3.1. Liver Surgery

Surgical site infections are a common cause of morbidity in hepatic resection without biliary tract reconstruction and occur in 5–20% of patients [9,10,11]. IDSA-ISI guidelines do not specifically address these procedures, but as they are clean-contaminated operations, the recommendation is a single preoperative dose of cefazolin. All this contrasts with Japanese Society of Chemotherapy (JSC) guidelines. Writers recommend starting antibiotics before surgery, redosing every 3 h during surgery, and continuing until 24 h after surgery [12]. In 2019, Takayama published a randomized non-inferiority trial comparing a one-day to a three-day regimen of antimicrobial prophylaxis in patients scheduled for open hepatic resection. As a primary outcome, SSI was diagnosed in 9.5% of patients in one day group and in 9.8% of patients in the day three group, meeting the non-inferiority hypothesis. Remote site infections (RSI) and drain-related infections were comparable too [13].

Shikawa retrospectively compared one-day vs. three-day prophylactic regimes in open and laparoscopic hepatic resections in the same year. In the propensity score-matched analysis, there was no difference in the incidence of post-operative complications between short- and long-term groups in open and laparoscopic surgery. The incidence rate of SSI was comparable between short- and long-term groups, with a far lower rate in laparoscopy (3.3% vs. 1.7%) than in open surgery (13.5% vs. 10.8%) [14]. In 2021, Xin Liew reported the results of an implementation program to increase adherence to antimicrobial prophylaxis guidelines. In the hepatectomy arm, they observed a significant increase in the compliance of a single cephazolin dose regimen without an increase in readmission rates. The authors stated that with using cefazolin only antimicrobial prophylaxis is safe and allows a significant decrease in antibiotic use. However, they are a surrogate outcome for post-operative infections, with no data about SSI [15]. The primary concern of eastern surgeons for the restrictive preoperative use of antibiotics is the high incidence in hepatic resection, about 40% in some trials [9,11], of massive blood loss. This is an indication for the supplementary administration of antibiotics [4].

Another difference between studies is the molecule used for prophylaxis. IDSA/ISI recommends cefazolin, cefoxitin, cefotetan, ceftriaxone, and ampicillin–sulbactam as prophylactic antibiotics for biliary tract procedures with a single 2 g dose of cefazolin as the regimen of choice [4]. The Japanese Society of Chemotherapy also recommends flomoxef, a cephamycin antibiotic used in some eastern studies [13,14]. In 2018, Starck published a retrospective matched case–control study comparing patients undergoing a hepatobiliary surgical procedure with and without an SSI. The population included 26% hepatic resections. Both in univariate and multivariate analysis, the broadening of the spectrum of antimicrobial prophylaxis against *Enterococcus* spp. and *Pseudomonas* spp. was not associated with a reduction in SSI, compared with cefazolin, cefazolin, and metronidazole or ampicillin/sulbactam [16]. This does not agree with a retrospective study published by Tang in 2018. He compared patients receiving a single ertapenem preoperative dose to patients treated with other prophylactic antibiotics (cefuroxime, cefoperazone, or piperacillin). After propensity score analysis in the ertapenem group, SSI incidence was significantly lower than in the non-Ertapenem group [17]. However, the incidence of SSI in the non-Ertapenem group (21.5%) was higher than those reported in other studies [13,14], suggesting a possible selection bias. Moreover, the alarming level of carbapenem resistance has presented particular challenges for managing a variety of infections [18].

Even though we cannot drive univocal conclusions from the analysis of the recent literature regarding the timing and drug for antimicrobial prophylaxis in hepatic resection, there is an interesting observation that we put forward. In recent works, especially for laparoscopic surgery, SSI rate is very low, up to 1.7% [14]. Factors associated with SSI in liver surgery include age, cancer stage, type of procedure, operation time, and blood loss [2]. Improvements in surgical techniques and enhanced recovery pathways [19] could have a more influential role than antimicrobics in reducing infective complications. This concludes a recent metanalysis published by Tao Guo in 2019 [20]. They conducted a network metanalysis of five eastern randomized trials comparing four antibiotic prophylaxis strategies: preoperative only, post-operative short (that is prolongation of prophylaxis for a maximum of 2 days), post-operative long (prolongation for more than two days post-operatively) and a negative control (no antibiotic prophylaxis). The metanalysis revealed that the application of no antibiotic exhibited the highest probability of achieving the lower rate of SSI. However, the authors observed that only one trial [10] reported no prophylaxis data and showed no significant difference in the antimicrobial arm. This fact implies that no direct statistical evidence supports the conclusion that no prophylaxis is the best strategy to lower post-operative infections.

Improvement in surgical technique and perioperative management lead to a very low SSI rate in hepatic resection without biliary reconstruction. One preoperative 2 g cefazolin dose may be adequate for surgical prophylaxis. Avoiding antimicrobial prophylaxis could be a topic for future research.

### 3.2. Antimicrobial Prophylaxis in Patients with Preoperative Biliary Drainage

IDSA/ISI guidelines consider pancreatoduodenectomy as a gastroduodenal procedure with the recommendation of a single dose of 2 g of cefazolin preoperatively for antibiotic prophylaxis [4]. However, patients with periampullary neoplasm often present with obstructive jaundice and cholangitis or undergo preoperative treatment. Their number is likely to increase because neoadjuvant treatment effectively improves resectability and survival in locally advanced pancreatic cancer [21]. In these cases, preoperative biliary drainage is necessary to recover from coagulopathy and immune dysfunction associated with hepatic impairment or increase treatment compliance. Stent placement, however, creates communication between the duodenum and biliary tree, facilitating bacterial migration and colonization. Current guidelines do not specifically address the clinical significance of bacterobilia, and there are no indications of possible modifications of prophylaxis according to bile culture results [4].

Preoperative biliary drainage has been demonstrated to increase the likelihood of positive intraoperative bile cultures, especially polymicrobial bile growing, with increased antibiotic resistance. In 2018, Hentzen published a retrospective multicentric study comparing patients undergoing pancreaticoduodenectomy (PD) who did or did not undergo preoperative drainage (PBD). Almost 90% of patients who underwent PBD (*n* = 175) had positive bile cultures (87.9%) compared with only 31.8% of non-drained patients (*p* < 0.001) [22]. In a retrospective work by Windisch, 37 patients operated on for periampullary tumors were included, 29 (78%) in the PBD group and 8 (22%) in the no biliary drainage group. In the PBD group, there was a significant increase in the positive bile culture compared to the undrained patients (*p* > 0.002) [23]. In a work by Sugimachi, a retrospective series of 51 patients who underwent PD for a malignant tumor revealed that bile culture was positive in 27 of 30 cases (90%) with preoperative biliary drainage and 1 of 21 cases (5%) without drainage [24].

Another important issue is the changes in the local bile microbiome induced by BD. Bacterobilia developed after drainage is often polymicrobial, changing toward colonization by enterococci and fungi [23]. In 2019, Kruger published a retrospective study including 285 patients with pancreatic head resection. Patients were divided into four groups according to the presence or absence of cholestasis and preoperative biliary drainage. Bacterobilia (BB) was more frequent in the subgroup with preoperative drainage (BB for PBD+: *n* = 120, 83.3% vs. BB for PBD-: *n* = 30, 21.4%; *p* < 0.01). Moreover, among patients with preoperative cholestasis, bacterobilia was more frequent in the subgroup with preoperative drainage. When analyzing microbiological data, a broad spectrum of bacteria and polymicrobial colonization was significantly characteristic of biliary drainage patients. They also found a more frequent detection of *Enterococcus* in the drainage group [25]. Bilgic, in 2020, published a retrospective study evaluating the effects of preoperative diagnostic and therapeutic biliary procedures on the development of SSI.

Ampicillin/sulbactam resistance was significantly more common in the PBD group (67% vs. 22%, *p* = 0.002). Meropenem, piperacillin-tazobactam, and ciprofloxacin resistance were also higher, although statistically not significant [26]. Additionally, in a work by Camman, a retrospective series of 243 patients with hepatobiliary surgery with biliodigestive anastomosis showed that stenting was associated with a higher rate of ampicillin (*p* = 0.091, OR = 1.72) and or ciprofloxacin-resistant bacteria (*p* < 0.001, OR = 3.48), resulting in 74.2% of all stented patients with a resistant bacterium in the bile [27]. Microbiological data are usually derived from intraoperative biliary culture, obtained after biliary tree transection. De Pastena, in a prospective series published in 2017, investigated the correlation between a rectal swab (RS) and intraoperative biliary culture as a possible antimicrobial stewardship strategy to guide surgical prophylaxis. RS culture showed a perfect correlation (species and phenotypic antibiotic susceptibility pattern) with bile culture in 157 patients (86.7% of cases), and preoperative biliary drain (PBD) was the single independent preoperative risk factor associated with RS positivity. This study suggests that the preoperative RS could be a promising strategy to predict the enteric colonization by MDR bacteria [28].

The correlation between bacterobilia (BB) and infective complications is less clear, especially surgical site infections. It is not clear whether SSI correlates with biliary stenting per se or if the subsequent development of bacterobilia, mainly MDR colonization, increases infectious risk. In the multicentric retrospective work by Fong including 1623 patients who underwent pancreatoduodenectomy from 3 high volume centers, preoperative biliary stenting was the strongest predictor of post-operative wound infection (odds ratio, 2.5; 95%CI, 1.58–3.88; *p* = 0.03), and there was a correspondence between microorganisms isolated in intraoperative bile cultures and those identified in wound cultures in patients with post-PD wound infections [29]. Bilgic et al. found a significant correlation between SSI rate and preoperative drainage. In 21% of cases, bile fluid and the surgical site presented similar bacterial species [26]. In a retrospective series by Sugimachi, incisional SSI correlated with multi-drug resistant (MDR) bacteria isolation, but not with organ/space SSI or overall post-operative complications [24].

Similarly, in the Sugawara retrospective series, the incidence of incisional, but not organ/space surgical site infection was significantly higher in patients with multidrug-resistant pathogen-positive bile culture compared to patients without MDR bacteria or with a negative culture [30]. They also found a correlation between MDR pathogens in preoperative bile culture and infectious complications confirmed in a subsequent publication by the same group in 2020 [31]. All patients, however, underwent preoperative drainage, either percutaneous or biliary nose. Costi et al., in a retrospectively analyzed population of 61 stented patients, found a correlation between *E. coli* isolation and poor outcome [32]. Gavazzi in 2016 published a retrospective evaluation of 180 patients who underwent PD and had intra-operative bile cultures. Stented patients had a significantly higher incidence of deep incisional surgical site infections (SSIs) (*p* = 0.038). *Enterococcus* spp. were the most frequent bacterial isolates in bile and all Enterococci tested were cefazolin resistant [33].

Conversely, in the multivariate analysis performed by Sugimachi on 69 patients who underwent pancreatoduodenectomy, bile culture was not statistically associated with SSI. Preoperative isolated microorganisms in bile were consistent with those detected in surgical sites only in 11 of 27 cases (41%) [24].

Based on the observation that preoperative drainage causes bacterobilia and can be associated with increased rates of SSI, some studies tested the hypothesis that preoperative bile-culture-targeted or at least upgraded antimicrobial prophylaxis can decrease post-operative infectious complication and SSI rate. We summarized the recent available literature in Table 1. Modified antimicrobial prophylaxis effectively decreases the SSI rate in all the papers we analyzed [27,34,35,36,37,38].

There is some criticism when interpreting these data.

The first is the rationale for the antibiotic prophylaxis change. It can be driven by local bile culture results [34,35,36,37] or be aimed at broadening the spectrum of efficacy, considering the available literature data [38]. In the Okamura clinical trial, patients were randomized to standard prophylaxis with cefazolin or a targeted group, which was administered antibiotics based on bile culture results. Patients in experimental arm experienced less SSI both in pancreaticoduodenectomy and hepatectomy (*p* = 0.001 and *p* = 0.025, respectively) [34]. As observed by Fong, who compared data coming from three high volume centers, however, there was marked institutional variation in the type of microorganisms cultured from both the intraoperative bile and wound infection cultures [29]; thus, single-center data cannot be generalized.

De Pastena [38] tested the efficacy of upgrading prophylaxis with piperacillin-tazobactam and found an improvement in hospital acquired infections and superficial SSI, also in extended spectrum beta-lactamase (ESBL) bacteria carriers. However, the current literature data do not entirely support the use of piperacillin-tazobactam in ESBL producing Enterobacteriaceae infections [39]. Moreover, this is not a randomized trial, and prophylaxis was the same for patients colonized and not colonized with MDR bacteria.

The second issue is the duration of antibiotic therapy, varying from 1 day [36,38] to all post-operative courses [27]. Even though some evidence exists that a short course of antibiotics perioperatively can reduce the overall rate of infectious complications after PD [40], IDSA/ISI guidelines recommend a single preoperative dose. In contrast, the Japanese Clinical Practice Guidelines for Antimicrobial Prophylaxis in Surgery [41] indicate a two-day course of antibiotics, supported by Sugawara RCT, including patients who underwent hepatic resection with bile duct reconstruction. They were randomized to 2 day (antibiotic treatment on days 1 and 2) or 4 day (on days 1 to 4) groups. Infectious complications and SSI were similar in the two groups [42].

Finally, there was only one prospective randomized trial [34]; other papers are retrospective series or prospective interventional non-randomized studies [27,35,36,37,38].

Even though it was not focused on antimicrobial prophylaxis, the FRAGERITA group study [43] offered an exciting perspective on this issue. This prospective study analyzed 312 patients from 5 European high-volume centers to evaluate the association between PBD duration and post-operative morbidity after pancreatoduodenectomy. The population study was stratified by stent duration in three groups: short (<4 weeks), intermediate (4–8 weeks), and long (>8 weeks). Patients with a stent duration of more than four weeks had the highest likelihood of bacterobilia and highest rates of MDR bacteria detection, but the morbidity rates were lower than that of the short group. It could be explained by the improvement in host immune conditions and remodulation of the biliary microbiome, reducing the pathogenic activity of MDR bacteria. Moreover, patients were administered standard prophylaxis with cefazolin 2 g or cefoxitin 2 g plus metronidazole 500.

From this study, we can derive that patients’ homeostasis rather than antimicrobial prophylaxis plays a paramount role in reducing post-operative morbidity. Moreover, the time from biliary drainage could be an essential element in decision making for surgical prophylaxis.

### 3.3. Low-Risk Cholecystectomy

According to IDSA/SIS/SHEA guidelines [4], a single dose of cefazolin should be administered in patients undergoing open biliary tract procedures, and antimicrobial prophylaxis is not necessary for low-risk patients undergoing elective laparoscopic cholecystectomy. Risk factors include diabetes, an anticipated procedure duration exceeding 120 min, the risk of intraoperative gallbladder rupture, age > 70 years, the risk of conversion to open, ASA classification ≥ 3, biliary colic within 30 days, pregnancy, nonfunctioning gallbladder, and immunosuppression. Because some of these risk factors cannot be determined before the surgical intervention, they conclude that giving a single prophylaxis dose can be reasonable to all patients.

SSI rates in laparoscopic cholecystectomy are 0.5–1.5%, comparable to the expected infection rate of clean cases and far lower than open cholecystectomy [44]. In a retrospective review of an extensive U.S. national database, all patient-related factors indicated in the current guidelines were significantly associated with SSI [45]. The role of high-risk features is less clear based on bile spillage or bacterial colonization. In the prospective series of Usuba, patients were divided into two groups: with or without intraoperative gallbladder perforation. SSI rates were significantly higher in patients with perforation, but they observed no difference in length of post-operative stay [46]. However, in this subgroup of patients, a single antibiotic dose effectively reduces the SSI rate [47,48].

Analyzing the available literature from 2015, we found five metanalyses [49,50,51,52,53] and eight clinical trials [54,55,56,57,58,59,60], including two RCT dealing with antibiotic prophylaxis in low-risk cholecystectomies. Evidence obtained from the metanalysis is summarized in Table 2. The results contrast with some works supporting antibiotics to reduce SSI [49,52,53], while others do not [50,51].

This is probably due to methodological bias. The first is prophylaxis definition and patient selection. The studies analyzed indeed were performed in different countries with different life environments and healthcare systems. First, one metanalysis [50] included only studies with a single preoperative dose administration as stated in WHO guidelines [5]. The other included studies compared no antibiotic with perioperative antibiotic administration up to 10 post-operative doses, which is a therapy rather than primary prophylaxis [49,51,52,53]. One study on acute cholecystectomy was also included in one case [52]. However, the most critical issue is the small sample size of available trials. They are thus underpowered to detect a significant difference since the SSI rate in laparoscopic cholecystectomy is very low (0.5–1%). In Matsui’s most extensive randomized trial [61], 518 patients were assigned to the antibiotics group and 519 to the no antibiotics group. The same author, in a subsequent metanalysis [51], stated that to reach a definite conclusion, a sample size of around 4500 cases with alfa error of 0.5 and power 0.8 is needed, based on an incidence of SSI of 2.1% in the antibiotic group and 3.1% in no antibiotic.

It is not easy to draw definitive conclusions about the effect of antimicrobial prophylaxis in low-risk cholecystectomy. Data from the literature are inconsistent, and some risk factors cannot be predicted before surgery. Thus, in our opinion, a strict preoperative cefazolin single dose strategy can be reasonable until a large-scale, multicentric RCT brings definitive conclusions.

## 4. Discussion

HPB surgery includes a wide range of surgical procedures. According to the current guidelines, a single preoperative dose of cefazolin is the gold standard for antimicrobial prophylaxis. However, there are at least three unsolved issues.

The first one is antimicrobial prophylaxis in hepatic resections without bile duct reconstruction. According to the recent literature, the risk of SSI is very low, especially for laparoscopic procedures. For this reason, a single cefazolin dose is effective. Further research will clarify if no antibiotic at all is a better strategy.

A second issue, not addressed in the current guidelines, is whether patients with obstructive jaundice who undergo preoperative drainage should receive antimicrobial prophylaxis based on microbiological biliary samples. Preoperative drainage increases the risk of bacterobilia and induces changes toward colonization by enterococci and fungi with increased antimicrobial resistance. According to the current literature, a targeted strategy is more effective than cefazolin in decreasing SSI. Thus, in patients who undergo surgical intervention within four weeks after biliary drainage, a single dose of piperacillin-tazobactam may be more effective than cefazolin in reducing SSI. However, we need more robust evidence from prospective randomized trials.

Finally, in low-risk cholecystectomy, even though the SSI rate is comparable to a clean procedure, it is not clear whether we can omit antimicrobial prophylaxis. Different metanalyses show contrasting results. This is probably due to the inadequate sample size and different schedules in the included trials. Thus, in our opinion, a strict one preoperative cefazolin single dose strategy can be reasonable until a large-scale, multicentric RCT brings definitive conclusions.

The main limitation of this paper is the absence of a metanalysis of data, especially for antimicrobial prophylaxis in patients with preoperative drainage.

We can conclude that a single preoperative dose of cefazolin for HBP surgery is indicated for antimicrobial prophylaxis. For patients with obstructive jaundice who undergo a surgical operation within 4 weeks from biliary drainage, a shift towards piperacillin-tazobactam may be considered. Further research is needed to clarify whether antimicrobial prophylaxis can be omitted in hepatic resection without reconstructing the biliary tree and low-risk cholecystectomies.

## Figures and Tables

**Table 1 antibiotics-11-00194-t001:** Patients underwent HBP surgery.

Author	Year	Methods	Methods	Conclusions
Okamura	2017	Prospective rand.	Patients who underwent HPB cancer surgery with biliary reconstruction. Before surgery, subjects were randomly allocated to a: - **target group**-administered antibiotics based on bile culture results - **standard group**-administered cefmetazole *Administration of antibiotic agents was continued until POD 2.*	The frequency of SSI after surgery was significantly lower in the targeted group than in the standard group.
Sano	2018	Retrospective	Pancreatoduodenectomy patients who underwent endoscopic biliary stenting. - **Cefazolin sodium hydrate** was administered as perioperative prophylactic antibiotic therapy from 2010 to 2014; - **Ceftriaxone** was administered from 2014 to 2017 based on the results of institutional culture surveillance. *Administration of antibiotic agents was continued until POD 2.*	The overall surgical site infection incidence in the Ceftriaxone group was significantly lower than that in the Cefazolin sodium hydrate group for Clavien-Dindo grade ≥ II.
Tanaka	2018	Prospective non random.	Patients who underwent pancreaticoduodenectomy. - **Cefmetazole** (which is routinely administered according to the CDC guidelines for Class II surgical wounds), - **VCM + PIPC/TAZ** (sensitive to the most commonly detected species in preoperative bile culture and postoperative infection culture of the ward). *In the CMZ group, after surgery (on the operation day), 1 g of CMZ was performed on the operation day only.* *In the VCM + PIPC/TAZ group, after surgery 4,5 g of PIPC/TAZ and VCM was continued every 8h until POD 1.*	The frequency of SSIs was significantly lower in the VCM + PIPC/TAZ group than in the cefmetazole group. Postoperatively, significantly fewer patients in the VCM + PIPC/TAZ group required ≥ 15 days of additional antibiotic administration than those in the cefmetazole group.
Cengiz	2019	Prospective non random.	Patients who underwent pancreaticoduodenectomy. - **cefalexin** - **ceftriaxone and metronidazole** (after evaluation of the local antibiogram) *A single dose of antibiotic was administered within one hour of the incision.*	A change in antibiotic prophylaxis prior to PD based on the local microflora resulted in reductions in SSI, POPF, and Clostridium difficile rates.
Cammann	2016	Retrospective	Patients who underwent hepatobiliary surgery with biliary reconstruction by BDA. - **ampicillin**/**sulbactam** - **ciprofloxacin** (according to the institutional guidelines) *A change of the antibiotic regime was made according to the results of the bile culture or in case of infectious complications in the postoperative course.*	Patients from the ciprofloxacin group had an increased risk of postoperative cholangitis than patients treated with ampicillin/sulbactam.

**Table 2 antibiotics-11-00194-t002:** Evidence obtained from the metanalysis.

Author	*n* of Studies	Primary Outcome	Results	Conclusion
Bo Liang 2016 [49]	21 RCT with 5207 patients	SSI and global infection	Antibiotics significantly reduce SSI (*p* = 0.001) and global infections (*p* = 0.001)	Support the use of antibiotics
Gomez-Ospina [50]	18 studies with 4087 patients	SSI	No difference in SSI with RD*of -0.00 (95% CI#-0.001 TO 0.001)	Antibiotics are not necessary
Pasquali 2016 [52]	19 studies with 5259 patients	SSI, distant infections, overall nosocomial infection and adverse reactions to antibiotics	No significant difference in SSI (*p* = 0.21) and distant infections (*p* = 0.06)	Antibiotics are not necessary.
Matsui 2018 [51]	Systematic review of 7 metanalysis	SSI, distant and overall infection	Antibiotics significantly reduce the risk of SSI (RR§ 0.71), distant (RR 0.37) and overall infection (RR 0.50)	Support the use of antibiotics
Kim 2018 [53]	28 RCTs, 3 prospective studies, and 3 retrospective Studies with 12121 patients	SSI, superficial SSI, deep SSI	prophylactic antibiotics were not effective in preventing deep SSI (*p* = 0.98) but effective in reducing SSI (*p* = 0.003) and superficial SSI (*p* = 0.002)	Support the use of antibiotics

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
