# Peer review of "Antibiotic Prophylaxis for Hepato-Biliopancreatic Surgery—A Systematic Review"

_antibiotics, 2022, doi:10.3390/antibiotics11020194_

Round 1

Reviewer 1 Report

The manuscript entitled “Antibiotic prophylaxis for hepato-bilio-pancreatic surgery. A systematic review” by Francesca Steccanella, et al.  aims to systematically review literature to assess the role of antimicrobial prophylaxis (AP) in HPB elective surgery.

The manuscript is well organized and provides a nice summary of the existing literature regarding this topic.

There are some points that need to be addressed:

The authors mentioned in their methodology that “Articles published in English between 2015 and 2021 were retained. Studies published before 2015 were not included in this review because they preceded the WHO guidelines on SSI prevention”. Despite this, I found many references citing articles published before 2015, why is that?

The manuscript requires extensive English revision. There are many grammatical issues and many sentences that could be significantly improved.

The manuscript is lacking a Discussion section, with a summary of evidence and interpretation. The authors also should include a section with limitations. The manuscript is also lacking a conclusion.

Author Response

  • We thank the first reviewer for his kind comment on our manuscript
  • We included in the three key points of the narrative review only papers from 2015. Other references are in support of introductory concept (eg: “Surgical site infections (SSIs) are one of the most important determinants of morbidity after hepatobiliary surgery. Their frequency after HPB surgery is highly variable, depending on the complexity of the procedure and intrinsic characteristic of the patient. It can vary from 1-2% after elective cholecystectomy to 25% after pancreatoduodenectomy)
  • We included a Discussion section, with a summary of evidence and interpretation, a section with limitations and a conclusion.
  • We reviewed the English translation as required

Reviewer 2 Report

Although this is an excellent review of antibiotic prophylaxis in HPB surgery, I do not agree with the conclusion that Cefazolin should be used in all HPB patients.

After duodenopancreatectomy, patients can sometimes present several weeks later with biliary sepsis in the setting of no intra-abdominal collections (leaks). This can often be treated with IV antibiosis. Because of this authors have begun using IV piperacillin-tazobactam prophylactically. Often times for 48 hours or longer if a persistent leucocytosis exists in the immediate post-operative period. Although the relevant literature is cited, there is a problem with the numbering of the bibliography. 

Because of this numbering issue I could not verify where this manuscript is cited.

De Pastena M, Paiella S, Azzini AM, Zaffagnini A, Scarlini L, Montagnini G, Maruccio M, Filippini C, Romeo F, Mazzariol A, Cascio GL, Bazaj A, Secchettin E, Bassi C, Salvia R. Antibiotic Prophylaxis with Piperacillin-Tazobactam Reduces Post-Operative Infectious Complication after Pancreatic Surgery: An Interventional, Non-Randomized Study. Surg Infect (Larchmt). 2021 Jun;22(5):536-542. doi: 10.1089/sur.2020.260. Epub 2020 Oct 23. PMID: 33095107.

Please rectify, revise and resubmit.

In the US, this is currently being studied.

https://clinicaltrials.gov/ct2/show/NCT03269994

The sponsor is Memorial Sloan Kettering. Please reference this study if not already done.

Author Response

  • We thank the second reviewer for his kind comment on our manuscript
  • We reviewed numeration of references and we found no mistake. Also, the work by De Pastena et al. was included. We conclude that in patients with preoperative drainage one can consider using piperacillin-tazobactam as a prophylaxis, especially if surgical operation is performed within 4 weeks. We think however that evidence in literature is not so strong.